# The NFκB-inducing kinase is essential for the developmental programming of skin-resident and IL-17-producing γδ T cells

**Florian Mair[1], Stefanie Joller[1], Romy Hoeppli[1‡], Lucas Onder[2], Matthias Hahn[3], Burkhard Ludewig[2], Ari Waisman[3†], Burkhard Becher[1*†]**

[1]Institute of Experimental Immunology, University of Zurich, Zurich, Switzerland; [2]Institute of Immunobiology, Kantonsspital St. Gallen, St. Gallen, Switzerland; [3]Institute for Molecular Medicine, University Medical Center, Johannes-Gutenberg University of Mainz, Mainz, Germany

**Abstract** γδ T cells contribute to first line immune defense, particularly through their ability for rapid production of proinflammatory cytokines. The cytokine profile of γδ T cells is hard-wired already during thymic development. Yet, the molecular pathways underlying this phenomenon are incompletely understood. Here we show that signaling via the NFκB-inducing kinase (NIK) is essential for the formation of a fully functional γδ T cell compartment. In the absence of NIK, development of Vγ5+ dendritic epidermal T cells (DETCs) was halted in the embryonic thymus, and impaired NIK function caused a selective loss of IL-17 expression by γδ T cells. Using a novel conditional mutant of NIK, we could show in vivo that NIK signaling in thymic epithelial cells is essential for the thymic hardwiring of γδ T cell cytokine production.

*For correspondence: becher@immunology.uzh.ch

†These authors contributed equally to this work

Present address: ‡Department of Surgery, Child and Family Research Institute, University of British Columbia, Vancouver, Canada

Competing interests: The authors declare that no competing interests exist.

## Introduction

γδ T cells, together with αβ T cells and B cells, are the only cells in mammals capable of generating diverse antigenic receptors by somatic gene rearrangement, which enables them to specifically recognize and respond to a vast array of antigens. However, while the essential role of αβ T cells and B cells for the induction of long-lasting protective immune responses is undisputed, γδ T cells due to their low abundance in peripheral lymphoid organs have been often ignored by the scientific community (*Allison and Havran, 1991*). This has changed in the past decade, as it became evident that γδ T cells are important players not only during protective immune reactions (*Lockhart et al., 2006*), but also in various models of autoimmune disease (*Petermann et al., 2010*; *Cai et al., 2011*; *Pantelyushin et al., 2012*).

Several features distinguish γδ T cells from their αβ counterparts (reviewed in *Vantourout and Hayday, 2013*). Most notably, γδ T cells recognize qualitatively different antigens. The scope of the potential antigens has not been revealed entirely, but known targets include non-classical MHC molecules (*Strid et al., 2008*), phosphoantigens (*Constant et al., 1994*) as well as a discrete set of non-self proteins (*Zeng et al., 2012*). The second main characteristic of γδ T cells is their ability for rapid production of pro-inflammatory cytokines such as INF-γ and IL-17 independent of TCR engagement (*Sutton et al., 2009*). This feature has been suggested to be hardwired in developing γδ T cells already during thymic development, probably in dependency of self-antigen recognition (*Jensen et al., 2008*). The surface molecule CD27, which belongs to the tumor necrosis (TNF) receptor superfamily has been shown to distinguish IFN-γ-producing from IL-17-producing γδ T cells (*Ribot et al., 2009*), as does the differential expression of the transcription factors T-bet and RORγt and certain chemokine receptors such as CCR6 (*Haas et al., 2009*).

**eLife digest** Our bodies are protected from infection and disease by several different types of immune cells. Gamma delta T cells are unusual in that they only make up a small proportion of the immune cells of the body, yet are present in many different animal species. These peculiar T cells are primarily found in the tissues that line the body (such as the skin, lung and gut) and are part of the first stage of the immune response that occurs when an invading microbe enters the body.

Gamma delta T cells, like all other T cells, develop in an organ called the thymus, which is found in the chest. Although several complex signaling pathways have been identified that help specific immune cells to develop, there are still many open questions about them. It is also unclear if other cells in the thymus influence how gamma delta T cells develop.

Mair et al. engineered mouse embryos that could not produce a signaling molecule known as NIK in certain subsets of their cells. This revealed that NIK is important for a structural cell in the thymus to instruct the early stages of gamma delta T cell development. However, gamma delta T cells themselves do not need to be able to produce NIK or the signaling pathway that it triggers.

Further work will focus on discovering the exact way in which the structural cells of the thymus interact with gamma delta T cells. This will help us understand better how developing immune cells are 'educated' in the thymus so that they are able to work effectively in the adult.

Although the genetic diversity of the TCR $\gamma\delta$ locus would theoretically allow an even higher number of different TCR specificities than for $\alpha\beta$ T cells (*Bonneville et al., 2010*), $\gamma\delta$ T cells show a remarkable limitation in their TCR variability. Moreover, $\gamma\delta$ T cells with a given TCR specificity often home to non-lymphoid organs, where the majority of mature $\gamma\delta$ T cells can be found. The prime example of tissue homing-behavior is observed in dendritic epidermal T cells (DETCs), which are found in the murine epidermis and almost exclusively express a clonotypic TCR consisting of the V$\gamma$5 and V$\delta$1 chains (*Asarnow et al., 1988*). V$\gamma$5$^+$ DETCs are among the very first T cells to develop during embryogenesis, populating the epidermis already around day 15/16 of fetal development (*Xiong et al., 2004*). Also other subsets of $\gamma\delta$ T cells, such as V$\gamma$4$^+$ $\gamma\delta$ T cells (mostly IL-17 producers), which migrate predominantly to the dermis and lung tissue, have been shown to develop only early in ontogeny (*Allison and Havran, 1991*; *Haas et al., 2012*; *Prinz et al., 2013*).

The underpinning signaling pathways and precise developmental program as well as the mechanisms underlying the pre-determined cytokine profile of $\gamma\delta$ T cells have been only partially resolved. One major step forward was the identification of Skint-1 (Selection and upkeep of intraepithelial T cells-1), which has an essential function for the development of V$\gamma$5$^+$ DETCs (*Lockhart et al., 2006*; *Lewis et al., 2006*; *Boyden et al., 2008*). Expression of Skint-1 by medullary thymic epithelial cells (mTECs) (*Barbee et al., 2011*) leads to a signaling cascade in developing $\gamma\delta$ thymocytes that induces the typical gene expression profile of DETCs, concomitant to their preferential homing to the skin (*Turchinovich and Hayday, 2011*; *Vantourout and Hayday, 2013*). On the other hand, the induction of the typical gene signature of IL-17-producing V$\gamma$4$^+$ $\gamma\delta$ T cells depends on the function of the transcription factor Sox13 (*Gray et al., 2013*), while the required receptors remain elusive.

It has been known for some time that signaling pathways leading to activation of NF$\kappa$B are essential for the function of various immune cell compartments (reviewed in *Li and Verma, 2002*). The so-called classical pathway of NF$\kappa$B activation is induced mainly by molecular danger signals such as LPS or certain cytokines such as TNF or IL-1$\beta$. Non-canonical NF$\kappa$B signaling, which requires the NF$\kappa$B-inducing kinase (hereafter referred to as NIK, the encoding gene being *Map3k14*,) (*Malinin et al., 1997*; *Yin, 2001*) is activated by a discrete subset of TNF family members, including CD40L, Lymphotoxin (LT), BAFF (B cell activating factor) and RANKL (Receptor activator of NF$\kappa$B ligand). Signaling via the non-canonical pathway has been shown to fulfill roles in B cell survival (*Sen, 2006*), activation of T cells (*Matsumoto et al., 2002*; *Ribot et al., 2009*), maturation and function of dendritic cells (DCs) (*Hofmann et al., 2011*) as well as the induction of tolerance (*Zhu et al., 2006*; *Bonneville et al., 2010*), but it has not been studied in the context of $\gamma\delta$ T cell biology in detail. Previous work has suggested that signaling via the CD70-CD27 axis (which belongs to the tumor necrosis factor receptor superfamily) can activate NF$\kappa$B p52 processing (*Ribot et al., 2010*). Also, LT$\beta$R signaling has been implicated in the differentiation program of bulk $\gamma\delta$ thymocytes (*Silva-*

*Santos, 2005*), and the NFκB family members RelA and RelB were shown to guide the development of IL-17-producing Vγ4⁺ γδ T cells (*Powolny-Budnicka et al., 2011*). Furthermore, the genesis of invariant NKT cells was claimed to depend on signaling via NIK (*Elewaut et al., 2003*).

In the present study we set out to systematically analyze the impact of non-canonical NFκB signaling on the development and function of γδ T cells. In line with previous studies that were using knockout models of the TNF receptor family member RANKL (*Roberts et al., 2012*) we found that NIK-deficient mice showed a drastic loss of DETCs in the epidermis. Of note, in the absence of NIK most of the IL-17 production by splenic and lung-resident γδ T cells was lost, while their ability to express IFN-γ was not affected. Mechanistically, this was accompanied by a loss of *Rorc* and *Sox13* expression in NIK-deficient CD27⁻ γδ T cells, while in turn expression of *Tbet* and *Egr3* was increased. While previous studies reported trans-conditioning of developing γδ T cell precursors by CD4⁺ thymocytes (*Silva-Santos, 2005*; *Powolny-Budnicka et al., 2011*), our data suggest that NIK signaling specifically in thymic epithelium is essential for shaping the cytokine profile of the γδ T cell compartment.

## Results

### In the absence of NIK the development of DETCs is halted in the embryonic thymus

Previous studies have shown that the development of DETCs is partially dependent on signaling via the RANK-RANKL axis (*Roberts et al., 2012*). In line with this, we observed a disturbed pool of DETCs in the epidermis of adult *Map3k14⁻ᐟ⁻* mice (*Yin, 2001*), with only 30-–50% of the γδ T cells present expressing the canonical Vγ5⁺ TCR (*Figure 1A*). Since DETCs are among the very first T cells to develop in ontogeny and populate the epidermis already prior to birth, we analyzed the epidermis of mouse embryos at day 19 post conception. Whereas there was already a prominent population of Vγ5⁺ DETCs present in WT controls, DETCs were virtually absent in the skin of NIK-deficient embryos (*Figure 1B,C*).

The absence of DETCs in the epidermis of *Map3k14⁻ᐟ⁻* embryos led us to speculate that NIK-deficient DETC precursors fail to develop in the embryonic thymus. To test this notion, we analyzed thymi from *Map3k14⁻ᐟ⁻* and heterozygous controls at embryonic day 19 for the presence of Vγ5⁺ thymocytes. Indeed, these cells were present in NIK-deficient thymi, albeit at reduced numbers and with a consistent reduction in staining intensity of the TCR (*Figure 1D*). In order to assess the maturation status of the developing Vγ5⁺ thymocytes, we evaluated the expression level of various molecules that have been associated with normal DETC development, such as CD45RB, CD122, CD24 and CD62L (*Lewis et al., 2006*). The expected upregulation of CD45RB and CD122, which is typical for developing DETCs was not found in *Map3k14⁻ᐟ⁻* embryos. In turn, the downregulation of CD24 and CD62L which normally coincides with DETC maturation was also reduced (*Figure 1E*). Similar observations with respect to the expression of CD45RB were obtained during the analysis of thymi isolated from E17 embryos (*Figure 1—figure supplement 1*). Taken together, the loss of NIK abrogates normal development of DETC precursors in the embryonic thymus, corroborating previous findings using knockout animals for *Tnfrsf11a* (*Roberts et al., 2012*).

### NIK-deficient lymphoid and non-lymphoid CD27⁻γδ T cells show a selective loss of IL-17 production

Based on the role of NIK in the formation of the epidermal DETC pool we assessed the contribution of non-canonical NFκB signaling to the development and function of other γδ T cell compartments in more detail. For comparison, throughout our study we included both targeted *Map3k14⁻ᐟ⁻* mice (*Yin, 2001*) as well as *aly/aly* animals, which harbor a point mutation in NIK, expressing a dysfunctional protein (*Shinkura et al., 1999*). Both in the spleen and the lung of these mutant animals, the total number of γδ T cells as well as the frequency of the two most prominent subclasses expressing the Vγ4⁺ and Vγ1.1⁺ TCR was unchanged. Also, the distribution of the CD27⁺ and CD27⁻ subsets of γδ T cells was indistinguishable between *Map3k14⁻ᐟ⁻* and heterozygous control animals (*Figure 2A–C*).

However, when we assessed the primary function of γδ T cells, which is the rapid production of pro-inflammatory cytokines, we noticed that after in vitro stimulation of bulk peripheral lymphoid γδ T cells their ability to express IL-17 was strongly diminished. In contrast, the frequency of both IFN-γ

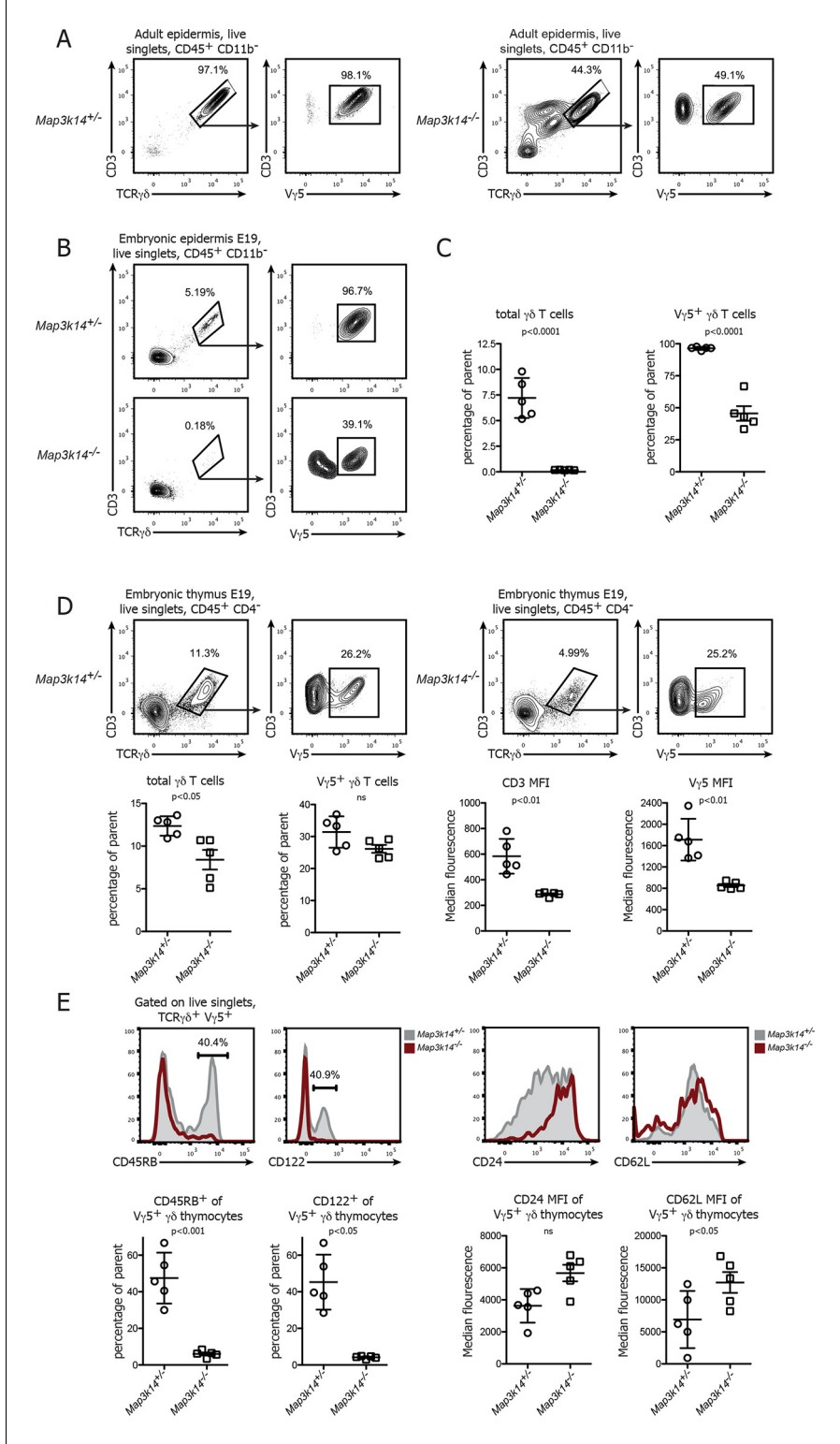

**Figure 1.** In the absence of NIK, the development of DETCs is blocked in the embryonic thymus. (**A**) Lymphocytes isolated from the epidermis of adult heterozygous control (left panel) and *Map3k14*[-/-] animals were analysed for the presence of Vγ5[+] DETCs. Pregating is on live singlets and CD45[+] CD11b[-] cells. (**B**) Analysis of the epidermal γδ T cell compartment of heterozygous control (upper panel) and *Map3k14*[-/-] embryos (day 19 post conception) after pregating on live singlets and CD45[+] CD11b[-] cells. (**C**) Summary of the frequency of total γδ T cells as well as Vγ5[+] cells within the γδ T cell gate. Data are mean +/- SD and are representative of two similar experiments. (**D**)

*Figure 1 continued on next page*

*Figure 1 continued*

Analysis of developing Vγ5$^+$γδ thymocytes in the thymi of E19 embryos. Flow Plots have been pregated on live singlets and CD45$^+$ CD4$^-$ cells. Lower panel depicts the summary of the frequency of total γδ thymocytes as well as Vγ5$^+$ cells within the γδ T cell gate in d19 embryonic thymi, and the median fluorescence intensity of the indicated markers. Data are mean +/- SD and representative of two similar experiments. (E) Analysis of the expression level of CD45RB, CD122, CD24 and CD62L on developing Vγ5$^+$γδ thymocytes isolated from E19 embryonic thymi. Grey shaded histograms depict heterozygous controls, red histograms *Map3k14$^{-/-}$* cells. Lower panel shows the summary for the frequency of positive cells for CD45RB and CD122 and the median fluorescence intensity of CD24 and CD62L, respectively. Data are mean +/- SD and are representative of two similar experiments.

The following figure supplement is available for figure 1:

**Figure supplement 1.** DETC development in NIK-deficient thymi at embryonic day 17.

---

and TNF-α-producing γδ T cells was similar between *Map3k14$^{-/-}$* and control animals (*Figure 2D,E*). For lung-resident γδ T cells, which include Vγ6$^+$ γδ T cells as one the most potent sources of IL-17, we also observed a reduction in the ability of NIK-deficient cells for IL-17 secretion, but this was accompanied by a slight increase in the frequency of IFN-γ production (*Figure 2F,G*). The ratio of Vγ4$^+$ to Vγ6$^+$ cells in the remaining fraction of IL-17-producing γδ T cells did not change (*Figure 2—figure supplement 1*). Similarly, for the steady state dermal γδ T cell compartment we observed that in the absence of NIK, γδ T cells were impaired in IL-17 expression (*Figure 2H*).

## NIK-deficiency leads to impaired *Rorc* and *Sox13* expression in CD27$^-$γδ T cells

To reveal the underlying mechanism for the loss of IL-17 in γδ T cells, we analyzed the expression of *Rorc* and *Il23r*, which are both necessary for normal IL-17 production (*Cua and Tato, 2010*; *Turchinovich and Hayday, 2011*). To this end, we purified CD27$^-$ and CD27$^+$ γδ T cells (*Figure 3A*) from *Map3k14$^{-/-}$* and heterozygous control animals and analyzed gene expression by qPCR. While in controls, CD27$^-$ γδ T cells expressed high levels of *Rorc* and *Il23r*, the same molecules were drastically reduced in NIK-deficient CD27$^-$ γδ T cells (*Figure 3B*), which was accompanied by an upregulation of *Tbet*. Furthermore, we assessed the expression of the transcription factors *Sox13* and *Egr3*, which have been proposed to be reciprocal regulators of γδ T17 cell development (*Turchinovich and Hayday, 2011*; *Gray et al., 2013*). While the expression level of both genes was not affected in CD27$^+$ γδ T cells, NIK-deficiency selectively in CD27$^-$ γδ T cells led to a loss of *Sox13* and a gain of *Egr3* expression (*Figure 3C*). Collectively, these data indicate that NIK signaling is essential for RORγt and as well as Sox13 expression in CD27$^-$ γδ T cells, thereby conferring their full potential for production of IL-17.

In order to assess the functional impact of NIK-lesioned γδ T cells, we treated *Map3k14$^{-/-}$* animals with Imiquimod, a well established system for the induction of psoriasis-like skin inflammation in mice (*van der Fits et al., 2009*; *Cai et al., 2011*; *Pantelyushin et al., 2012*). After treatment with Aldara (containing 5% Imiquimod), NIK-deficient animals displayed a slight, but not significant reduction of back skin thickness (*Figure 3D*). Analysis of the dermal γδ T cell compartment not only revealed reduced numbers of γδ T cells (data not shown), but also a significant reduction of both IL-17A and IL-17F expression by γδ T cells (*Figure 3E*). This suggests that NIK-signaling also in an inflammatory setting is essential for full IL-17 production by γδ T cells, but that other cell types, such as ILCs (*Pantelyushin et al., 2012*) can compensate this functional impairment during Aldara induced skin inflammation.

## Targeted deletion of NIK in mTECs, but not in γδ T cells causes abnormal DETC development

While our data and work from others (*Roberts et al., 2012*) clearly suggest a non-redundant role of non-canonical NFκB signaling in the development of DETCs, it has so far not been possible to determine whether NFκB signaling occurs in a γδ T cell intrinsic or extrinsic manner. To dissect the contribution of different cellular compartments *in vivo*, we generated a novel conditional mouse mutant that

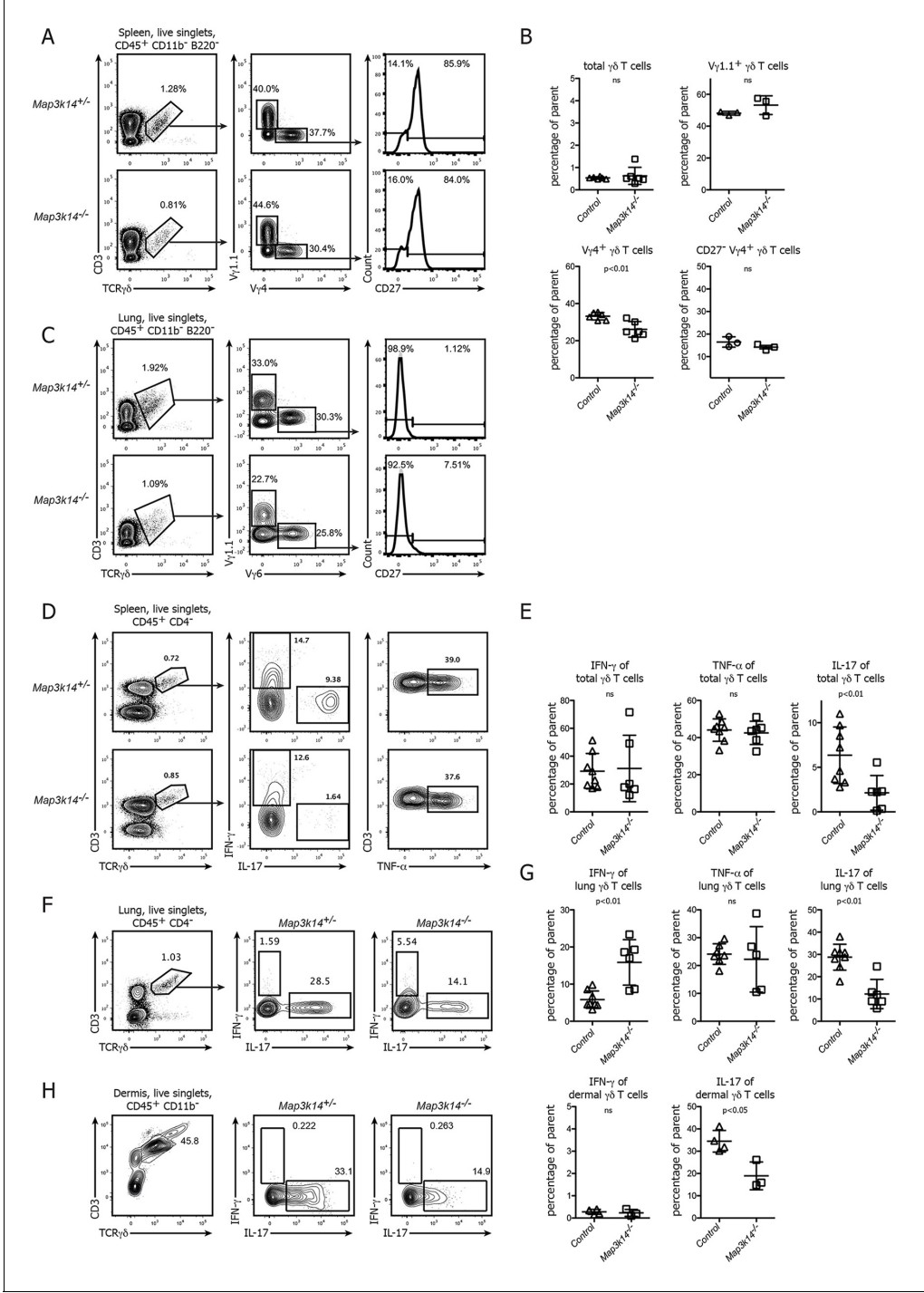

**Figure 2.** In NIK-deficient mice, γδ T cells selectively lose their ability for production of IL-17. (**A**) Flow cytometric analysis of the splenic γδ T cell compartment of heterozygous control (upper panel) and *Map3k14⁻/⁻* animals after pregating on live singlets and CD45⁺ CD11b⁻ B220⁻ cells. The distribution of Vγ1.1⁺, Vγ4⁺, and CD27⁻ Vγ4⁺ γδ T cells is shown. (**B**) Summary of two independent experiments. Data represent mean +/- SD of the indicated cell subsets. (**C**) Analysis of the lung-resident γδ T cell compartment of heterozygous control (upper panel) and *Map3k14⁻/⁻* animals after pregating on live singlets and CD45⁺ CD11b⁻ B220⁻ cells. The distribution of Vγ1.1⁺, Vγ6⁺, and CD27⁻ Vγ6⁺ γδ T cells is shown. (**D**) Flow cytometric analysis of IFN-γ, IL-17 and TNF-α expression by peripheral γδ T cells after PMA/Ionomycin stimulation, isolated from the spleen of heterozygous control (upper panel) and *Map3k14⁻/⁻* animals (lower panel). Left plots have been pregated on CD45⁺ CD11b⁻ CD4⁻ live singlets. (**E**) Summary of the frequency of cytokine producing cells within the γδ T cell compartment. Data represent mean +/- SD. (**F**) Flow cytometric analysis of IFN-γ and IL-17 expression by lung-resident γδ T cells isolated from the lung of heterozygous control (upper panel) and *Map3k14⁻/⁻* animals (lower panel). Left plots have been pregated on CD45⁺ CD11b⁻ CD4⁻ live singlets. (**G**) Summary of the frequency of cytokine producing cells within the lung-resident γδ T cell

*Figure 2 continued on next page*

Figure 2 continued

compartment. Data represent mean +/- SD. (H) Flow cytometric analysis of IFN-γ and IL-17 expression by dermal γδ T cells isolated from the skin of heterozygous control and *Map3k14*$^{-/-}$ animals. Right panel depicts the frequency of cytokine producing cells within the dermal γδ T cell compartment. Data show mean +/- SD and are representative of at least three independent experiments.

The following figure supplement is available for figure 2:

**Figure supplement 1.** Ratio of Vγ4$^+$ and Vγ6$^+$ cells in the lung-resident IL-17-producing γδ T cell compartment.

allows cell-type specific deletion of the *Map3k14* gene using the Cre/LoxP system. *Map3k14*$^{flox/wt}$ animals were generated by gene targeting in embryonic stem (ES) cells, thereby inserting LoxP sites up- and downstream of Exon 4 and 6 of the *Map3k14* gene locus, respectively (Supp. *Figure 1*). These mice were first crossed to a Deleter-Cre strain, followed by breeding the deleted locus to homozygosity. Examination of the resulting *Map3k14*$^{del/del}$ mice showed that these mice were lacking lymph nodes, suggesting that the conditional construct is working as expected (*Figure 4A*). To further verify the phenotype of *Map3k14*$^{del/del}$ mice, we compared the cytokine production of peripheral γδ T cells to WT and *Map3k14*$^{-/-}$ animals, showing that *Map3k14*$^{del/del}$ mice phenocopy the full knockout (*Figure 4B*). The same was true for the epidermal DETC pool (*Figure 4C*).

As a next step, we assessed whether the DETC defect observed in complete NIK-knockout animals is due to the requirement for NIK signaling in thymic epithelial cells. To specifically delete NIK in medullary thymic epithelial cells (mTECs), *Map3k14*$^{flox/flox}$ mice were crossed to *Ccl19-Cre* mice, which express Cre solely in lymph node stromal cells and mature medullary thymic epithelial cells (*Chai et al., 2013*; *Onder et al., 2015*). *Ccl19-Cre*$^+$ *Map3k14*$^{flox/flox}$ animals phenocopied the aberrant DETC development observed in germline *Map3k14*$^{-/-}$ mice, with a loss of Vγ5 expression by DETCs (*Figure 4D*). However, we observed some variation in the penetrance of the Ccl19-Cre transgene, with some litters presenting with a phenotype, while others did not, suggesting that the start of Cre expression driven by *Ccl19* promotor sharply coincides with the start of DETC development in the embryo. Thus, we decided to utilize an additional strain targeting the thymic epithelium, namely *Foxn1-Cre* mice in which Cre is under control of the *Foxn1* promotor and thus active in all thymic epithelial cells (both cortical and medullary) (*Gordon et al., 2007*). Corroborating our findings, *Foxn1-Cre*$^+$ *Map3k14*$^{flox/flox}$ showed a strong reduction in both frequency and number of skin-resident TCRγδ$^{high}$ DETCs (*Figure 4E*).

## NIK signaling specifically in thymic epithelial cells is essential for the formation of the IL-17-producing γδ T cell pool

To assess which cellular compartment requires NIK for the formation of IL-17 committed γδ T cells, we first deleted NIK specifically in CD27$^-$ γδ T cells using *Rorc-Cre*$^+$ *Map3k14*$^{flox/flox}$ animals. Analysis of Fatemap *Rorc-Cre*$^+$ *R26-Stop-flox-YFP* mice showed that around 90% of CD27$^-$ γδ T cells in the lung expressed yellow fluorescent protein (YFP), indicating a good targeting efficiency of this cellular subset (*Figure 5A*). However, lung-resident γδ T cells in *Rorc-Cre*$^+$ *Map3k14*$^{flox/flox}$ mice expressed IL-17 at levels comparable to wildtype controls (*Figure 5B*), suggesting that γδ T cell intrinsic signaling is dispensable for the commitment of IL-17-producing γδ T cells. To exclude a potential role of NIK signaling in the CD4$^+$ αβ thymocyte compartment (*Powolny-Budnicka et al., 2011*), we analyzed both peripheral and lung-resident γδ T cells in *CD4-Cre*$^+$ *Map3k14*$^{flox/flox}$ mice. Their ability for IL-17 production was comparable to wildtype controls (*Figure 5C*), demonstrating that also within the αβ thymocyte compartment NIK signaling is dispensable.

Based on these results we speculated whether loss of NIK signaling in TECs will impact on the commitment of IL-17-producing γδ T cells. To this end, we analyzed the lung-resident γδ T cell compartment of both *Ccl19-Cre*$^+$ *Map3k14*$^{flox/flox}$ and *Foxn1-Cre*$^+$ *Map3k14*$^{flox/flox}$ mice. While γδ T cells isolated from the lung of *Ccl19-Cre*$^+$ *Map3k14*$^{flox/flox}$ animals did not present with any abnormalities in cytokine production, γδ T cells isolated from P13 *Foxn1-Cre*$^+$ *Map3k14*$^{flox/flox}$ mice showed significant defects in their ability for IL-17 production both in the spleen and lung (*Figure 5D*), suggesting that indeed NIK signaling in thymic epithelial cells is needed for imprinting the cytokine commitment onto developing γδ T cells.

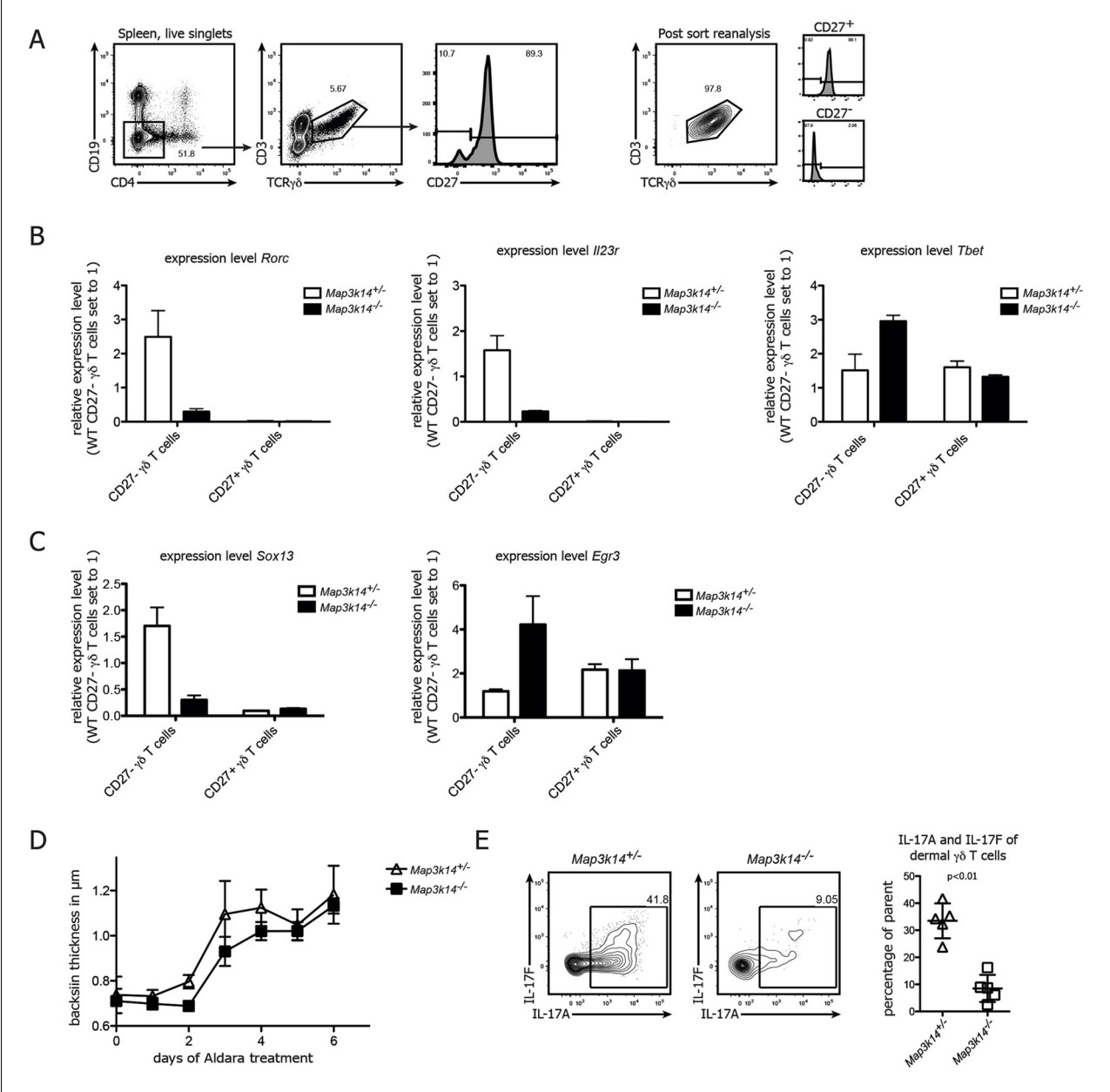

**Figure 3.** NIK-deficient CD27⁻ γδ T cells show reduced expression of *Rorc* and and *Sox13*. (**A**) Gating strategy used for sorting of CD27⁺ and CD27⁻γδ T cells. Right plot depicts post-sort reanalysis, which routinely yielded purities >95%. (**B**) Analysis of the expression level of *Rorc*, *Il23r* and *Tbet* mRNA in sorted CD27⁺ and CD27⁻ γδ T cells isolated from the spleen of the indicated genotypes. Data represent mean +/- SD. (**C**) Analysis of the expression level of *Sox13* and *Egr3* in sorted CD27⁺ and CD27⁻ splenic γδ T cells isolated from the indicated genotypes. (**D**) Backskin thickness after Aldara treatment of heterozygous control (open triangles) and *Map3k14⁻/⁻* animals (closed squares). Data are representative of two independent experiments. (**E**) IL17-A and IL17-F expression of dermal γδ T cell on day 6 after Aldara treatment.

## Discussion

In the past years, γδ T cells have attracted growing interest due to their unique features that place them at the interface of adaptive and innate immunity. Nevertheless, in contrast to αβ T cells, the signaling events underlying γδ T cell development and activation are much less understood (reviewed in *Prinz et al., 2013* and *Vantourout and Hayday, 2013*). Here, we have shown *in vivo* that NIK signaling in thymic epithelial cells is essential for the full function of two different γδ T cell populations, namely skin-resident DETCs as well as CD27⁻ IL-17-producing γδ T cells.

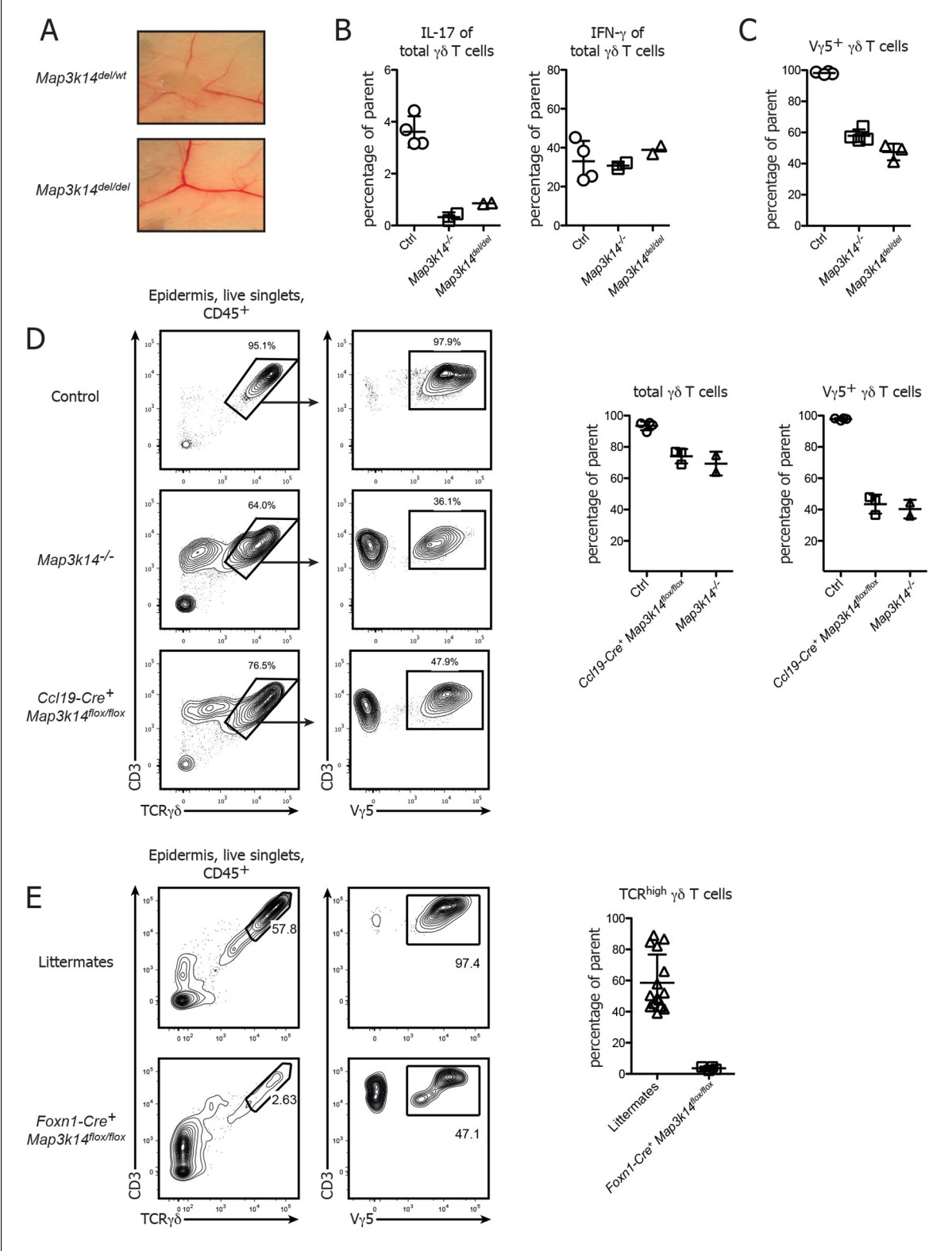

**Figure 4.** Conditional deletion of NIK in mTECs causes impaired DETC development. (**A**) *Map3k14^flox/wt^* mice have been crossed to Deleter Cre animals for deletion of the Loxp flanked gene locus. After subsequent breeding to homozygosity, *Map3k14^del/del^* mice were analyzed for the presence of lymph nodes. Upper panel depicts heterozygous controls. (**B**) Splenic γδ T cells from control, *Map3k14^-/-^* and *Map3k14^del/del^* mice were assessed for production of IL-17 (left panel) and IFN-γ (right panel) by intracellular cytokine staining. (**C**) The epidermal γδ T cell compartment of control, *Map3k14^-/-^* and *Map3k14^del/del^* animals was assessed for the frequency of Vγ5+ DETCs. (**D**) Lymphocytes isolated from the epidermis of adult control (upper panel),
*Figure 4 continued on next page*

Figure 4 continued

*Map3k14*[-/-] and *Ccl19-Cre*[+] *Map3k14*[flox/flox] animals were analysed for the presence of Vγ5[+] DETCs. Pregating is on live singlets and CD45[+] cells. Right panel depicts the summary of the frequency of total γδ T cells as well as Vγ5[+] γδ T cells in the epidermis of Control, *Map3k14*[-/-] and *Ccl19-Cre*[+] *Map3k14*[flox/flox] animals. Data represent mean +/- SD. (E) Frequency of total γδ T cells in the skin of 12–-13 day old control littermates and *Foxn1-Cre*[+] *Map3k14*[flox/flox] animals. Data represent mean +/- SD and is pooled from several independent experiments.

The following figure supplement is available for figure 4:

**Figure supplement 1.** Targeting strategy used for generation of the conditional *Map3k14*[flox/wt] strain.

Previous work has suggested an involvement of the NFκB-inducing kinase in the differentiation of both αβ and γδ T cells (*Eshima et al., 2014*). Contrary to the findings by Eshima and colleagues we could not observe a consistent overall reduction in the number of γδ T cells (except the DETC pool), yet we observed functional impairment of CD27[-] IL-17-producing γδ T cells. This discrepancy might result from two mutually non-exclusive explanations: first, *aly/aly* mice as well as *Map3k14*[-/-] mice due to their complex phenotype (including impairment of thymic negative selection) are prone to develop some degree of tissue inflammation over time, which is accompanied by an expansion of both CD4[+] and CD8[+] cells (*Tsubata et al., 1996*; *Ishimaru et al., 2006*; *Zhu et al., 2006*) and could thereby induce a relative decrease in the proportion of γδ T cells. Second, it has recently been established that the γδ T cell compartment cannot be fully reconstituted using bone-marrow chimeric animals (*Gray et al., 2011*; *Haas et al., 2012*). Using a genetic model that allows the cell-type specific deletion of NIK avoids all these confounding phenotypes of mice in which the gene locus of NIK (*Map3k14*) has been deleted in the germ-line.

Two members of the TNFR family, namely CD40 and RANK, have been implicated both in the development of DETCs and full maturation of mTECs (*Hikosaka et al., 2008*; *Akiyama et al., 2008*). In embryogenesis, these events occur at similar time points post conception, and it seems that in the embryonic thymus, developing DETC precursors signal via the RANKL-RANK axis to immature mTECs, inducing their full maturation to Aire[+] MHC-II[high] mTECs (*Roberts et al., 2012*). Hence, *Tnfrsf11a*[-/-] mice have reduced numbers of DETCs. Given that NIK is downstream of RANK (*Darnay et al., 1999*), one would expect a similar phenotype in *Map3k14*[-/-] animals, but the DETC deficiency that we observed in newborn *Map3k14*[-/-] mice was much more pronounced than in *Tnfrsf11a*[-/-] animals, suggesting that during DETC development several upstream receptors converge in their signaling to NIK, making it a key pathway during thymic DETC development. Importantly, our data obtained using the conditional NIK-mutant strongly suggests that for normal function of the γδ T cell compartment, NIK signaling does not act γδ cell-intrinsically but in trans by its function in thymic epithelial cells. Conditional deletion of NIK solely in thymic epithelial cells was sufficient to copy the DETC phenotype seen in complete germ-line NIK-knockout animals. We believe this to be the first genetic *in vivo* demonstration that non-canonical NFκB signaling is not required in DETCs in a cell-intrinsic manner, but through its activity in TECs.

The ability of CD27[-] γδ T cells to rapidly produce IL-17 even after being activated only by cytokines (*Sutton et al., 2009*; *Ribot et al., 2009*) has been the focus of intense research. Only the neonatal thymic microenvironment appears to have the capacity to imprint a stable cytokine profile to γδ T cells, since the IL-17-producing γδ T cell subsets cannot be reconstituted in adult animals by bone marrow transfer (*Haas et al., 2012*). Very recently, a hallmark paper has provided mechanistic insights into how this can be achieved: during embryonic development of certain γδ T cell subsets, the signaling threshold for activation of the TCR becomes markedly altered (*Wencker et al., 2013*). Hence, mature γδ T cells do not necessarily rely on a TCR stimulation as "'signal one"', although they can still respond to that (*Strid et al., 2011*).

However, while this intriguing mechanism (*Wencker et al., 2013*) explains the behavior of several γδ T cell subsets, the thymic cues orchestrating this hard-wiring of γδ T cells remain unclear. Previous work suggested either LTβR-dependent trans-conditioning by double positive thymocytes or γδ T cell intrinsic RelB activation (*Silva-Santos, 2005*; *Powolny-Budnicka et al., 2011*). Further work by Silva-Santos and colleagues emphasized the role of TNF-receptor family members such as CD27 as well as NFκBp52 activation (*Ribot et al., 2010*). However, the contribution of thymic stromal cells to these processes has not been experimentally addressed *in vivo*. Our data suggest that NFκB

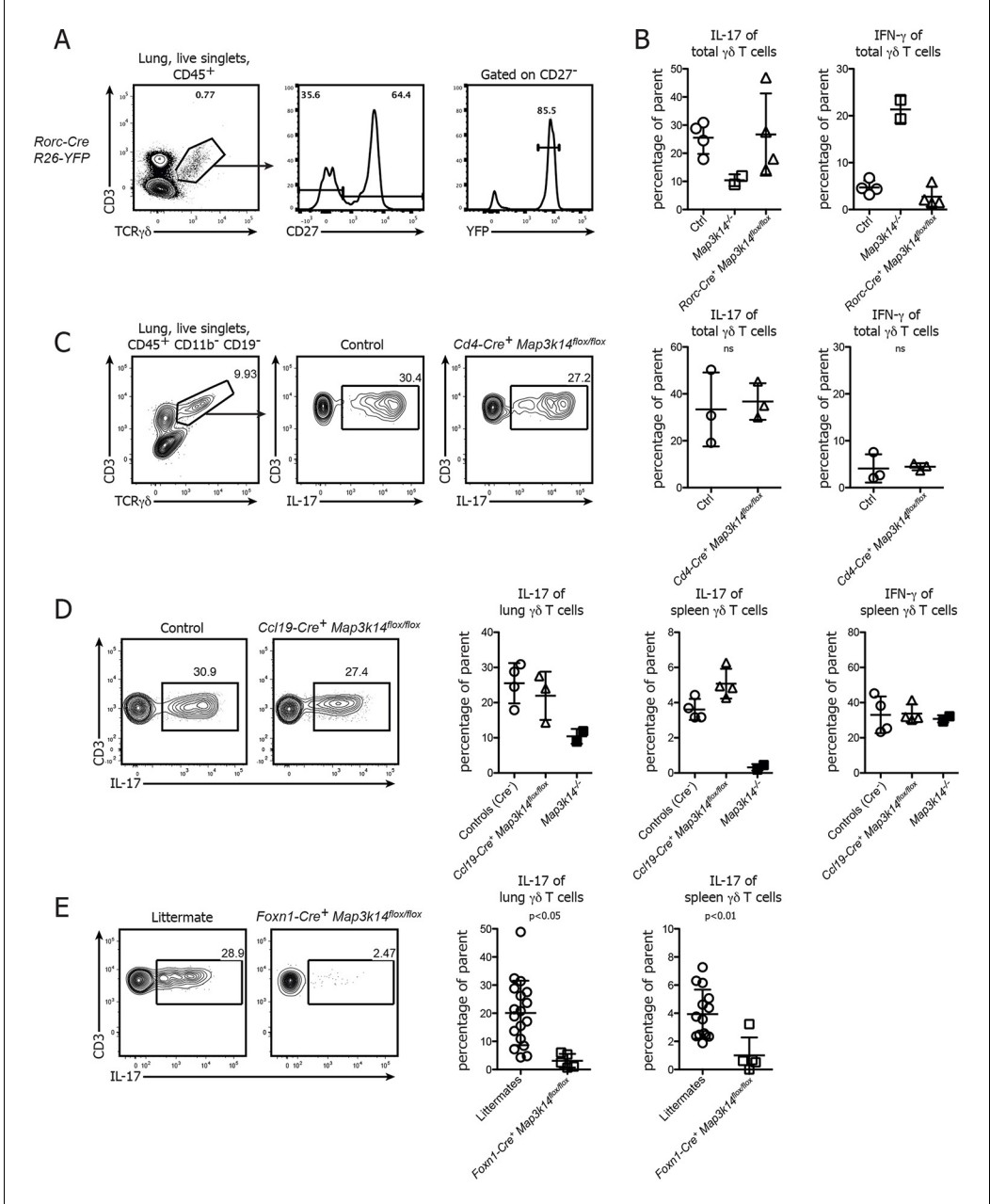

**Figure 5.** Conditional deletion of NIK in TECs causes not only a loss of DETCs, but also IL-17-producing γδ T cells. (**A**) γδ T cells were isolated from the lung of *Rorc-Cre⁺ R26-Stop-flox YFP*-mice and assessed for the frequency of YFP⁺ cells within the CD27⁻γδ T cell compartment. (**B**) Lung-resident γδ T cells from adult control, *Map3k14⁻/⁻* and *Rorc-Cre⁺ Map3k14^flox/flox* mice were assessed for production of IL-17 (left panel) and IFN-γ (right panel) by intracellular cytokine staining. (**C**) Lung-resident γδ T cells from adult control and *CD4-Cre⁺ Map3k14^flox/flox* mice were assessed for production of IL-17 (left panel) and IFN-γ (right panel). (**D**) Lung-resident as well as splenic γδ T cells from adult control and *Ccl19-Cre⁺ Map3k14^flox/flox* mice were assessed for production of IL-17 and IFN-γ. Data is representative of three independent experiments. (**E**) Frequency of IL-17-producing γδ T cells in the lung of 12-–13 day old control littermates and *Foxn1-Cre⁺ Map3k14^flox/flox* animals, plots have been pregated on live CD45⁺ singlets, CD3⁺ TCRγδ⁺ cells. Right panels depict the summary of several independent experiments for cytokine production in the lung (left) and spleen (right). Data are mean +/- SD.

signaling via NIK seems to be dispensable both within CD27⁻ γδ T cells as well as αβ T cells, at least for their ability for expression of pro-inflammatory IL-17. Of note, deletion of NIK in all thymic epithelial cells using *Foxn1-Cre* phenocopied the complete NIK-knockout in terms of IL-17 expression by CD27⁻γδ T cells, while this was not the case using *Ccl19-Cre*. The latter has been reported to target only maturing medullary thymic epithelial cells (*Onder et al., 2015*). This would suggest that the

instruction of γδ T cells is mediated by early or junctional TECs, or that the timing of *Ccl19-Cre*-mediated recombination sharply coincides with the expression of the γδ T cell instructing program in TECs. Based on our observation that NIK signaling in the early thymic stroma has a profound and long-lasting effect on the maturation and cytokine profile of developing γδ T cells, further studies are required in order to precisely define the nature of γδ T cell instructing molecules, and which thymic epithelial cell type is required for delivery. While our data does not exclude a role for canonical-NFκB signaling via RelA (*Powolny-Budnicka et al., 2011*), the impact of non-canonical signaling via NIK on the development of DETC precursors and IL-17-producing γδ T cells stems from signaling within the thymic stroma, for which the conditional NIK mutant is a unique tool for further studies. Future work will reveal whether NIK-deficient thymic stroma influences only γδ T cell development, or also the functional profile of the subsequently emerging αβ T cell compartment.

## Materials and methods

### Mice and animal experiments

C57Bl/6 (wildtype) mice were purchased from Jackson Laboratories and bred in-house at the Laboratory Animal Science Center (LASC) of the University of Zurich under specific pathogen-free (SPF) conditions. *Map3k14*[-/-] mice were kindly provided by Robert Schreiber (Washington University School of Medicine, USA), backcrossed to C57Bl/6 background and bred in-house. Alymphoplasia mice (here referred to as *aly/aly*) were obtained from Clea Laboratories (Japan) and bred in-house. *Map3k14*[flox/wt] mice were generated in collaboration with the lab of Ari Waisman (University of Mainz, Germany) by TaconicArtemis. Briefly, a targeting vector containing the Loxp-flanked Exons 4-–6 of the *Map3k14* loucs was introduced into embryonic stem (ES) cells, ES cell clones with homologous integration were selection and injected into blastocytes (*Figure 4—figure supplement 1*). *Ccl19-Cre* mice were generated in the lab of Burkhard Ludewig (Institute of Immunobiology, St. Gallen, Switzerland) by Lucas Onder (*Chai et al., 2013*). *Foxn1-Cre* mice were purchased from Jackson (*Gordon et al., 2007*). For the induction of psoriasis-like skin inflammation 55 mg of Aldara (containing 5% of Imiquimod) was applied to the shaved back skin of animals. Treatment of animals as well as measurement of skin thickness was done blinded. All animal experiments were approved by local authorities (Swiss cantonal veterinary office Zurich, KVET license numbers 86/2012, 70/2015, 100/2015 and 68/2013) and performed in strict accordance with the corresponding license.

### Lymphocyte isolation from the skin and lung

Unless stated otherwise, ears were taken as the donor organ. Ears were split into dorsal and ventral sides using forceps, followed by incubation with Dispase (2.4 mg/ml, Gibco) in HBSS with $Ca^{2+}$/ $Mg^{2+}$ at 37°C for 90 min Epidermal sheets were removed from the dermis, cut into small pieces using scissors and further digested using Collagenase Type 4 (0.4 mg/ml, Sigma) in HBSS with $Ca^{2+}$/ $Mg^{2+}$ and 10% FCS at 37°C for 60-–90 min The resulting suspension was mechanically disrupted using 19 g needles and a syringe. After filtering, the cell suspension was used in downstream applications.

Lungs were isolated, cut into small pieces using scissors and further digested using Collagenase D (0.5 mg/ml, Roche) in RPMI with 2% FCS and 25 mM HEPES. The resulting suspension was mechanically disrupted using 19 g needles and a syringe, followed by filtering and downstream analysis.

### Lymphocyte isolation from embryonic thymi

After setting up timed matings o/n, female mice were observed for signs of pregnancy, sacrificied on the indicated time points and embryos were isolated and euthanized by decapitation. The day after timed mating was considered day 0. Embryonic thymi were collected using fine forceps and a stereo microscope (Leica), and then dissociated by mechanical disruption using a cell strainer and a syringe pistil.

### Flow cytometry

Flow cytometric analysis was performed following standard methods (reviewed in *Perfetto et al., 2004*). All flourochrome-conjugated antibodies used were obtained either from BD (NJ, USA),

BioLegend (CA, USA) or eBioscience (CA, USA). In all stainings, dead cells were excluded using an Aqua or Near-IR Live/Dead fixable staining reagent (Life Technologies, now Thermo Fisher Scientific, MA, USA / BioLegend), and doublets were excluded by FSC-A vs FSC-H gating. For intracellular cytokine staining, cells were incubated 4 hours in IMDM containing 10% FCS with PMA (50 ng/ml) / Ionomycin (500 ng/ml) and GolgiPlug (Brefeldin A, BD). Cytofix/Cytoperm (BD) was used according to the manufacturers instructions, and Perm/Wash buffer was prepared in the lab (PBS containing 0.5% Saponin and 5% BSA). Analysis was performed using a LSR II Fortessa (special order research product, BD, equipped with 405 nm, 488 nm, 561 nm and 640 nm laser lines), cell sorting was carried out using a FACSAria III (BD). Data analysis was performed using FlowJo V9.x and 10.0.x (Treestar, OR, USA).

## qPCR analysis

Indicated cellular subsets were sort purified and collected in Trizol reagent. Afterwards, RNA was isolated using PureLink RNA Micro Kit (Thermo Fisher Scientific) following the manufacturers instructions. The eluted RNA was transcribed into cDNA using M-MLV reverse transcriptase (Thermo Fisher Scientific) or Superscript II (Thermo Fisher Scientific) and random primers. cDNA was diluted either 1:5 or 1:10 and used for qPCR with iTaq Universal SYBR Green Supermix (BioRad, CA, USA) and a BioRad C1000Touch / CFX384 real-time system according to the manufacturers instructions. The used primers (intron-spanning) were as follows:

*PolR2a* (145 bp): CTGGTCCTTCGAATCCGCATC
GCTCGATACCCTGCAGGGTCA
*Map3k14* (164 bp): CGAAACTGAGGACAACGAG
CACACTGGAAGCCTGTCTG
*Skint-1* (143 bp): TTCAGATGGTCACAGCAAGC
GAACCAGCGAATCTCCATGT
*Rorc* (114 bp): CATATGCCTCTCTGACAGAC
AAAAGAGGTTGGTGCGCTG
*Tbet*: CAACAACCCCTTTGCCAAAG
TCCCCCAAGCAGTTGACAGT
*Egr3*: CAACGACATGGGCTCCATTC
GGCCTTGATGGTCTCCAGTG
*Sox13*: GCTTTACCTATTCAGCCCAT
ACCTCTTCACCACAGGGG

## Statistical analysis

Unless stated otherwise, data is displayed as mean +/- standard deviation.

Statistical calculations were performed using GraphPad Prism, and significance was calculated using an unpaired t-test unless stated otherwise.

## Acknowledgements

We thank Robert Tigelaar (Yale School of Medicine, USA) for the kind donation of the 17D1 hybridoma, members of the Becher laboratory for helpful discussions and the Flow Cytometry Facility of the University of Zurich for cell sorting support. The study was financed through grants from the Swiss National Science Foundation (316030_150768, 310030_146130, CRSII3_136203) (all to BB), the Swiss Cancer League (BB), European Union FP7 projects TargetBrain (BB), NeuroKine (AW and BB) and ATECT (BB) and the university research priority project translational cancer research (BB). FM was supported by a fellowship (Forschungskredit) from the University of Zurich.

## Additional information

### Funding

| Funder | Author |
| --- | --- |
| Schweizerischer Nationalfonds zur Förderung der Wissenschaftlichen Forschung (SNF) | Florian Mair<br>Stefanie Joller<br>Romy Hoeppli<br>Burkhard Becher |
| European Commission | Florian Mair<br>Stefanie Joller<br>Romy Hoeppli<br>Burkhard Becher |
| Swiss Cancer League | Burkhard Becher |
| University of Zurich | Florian Mair<br>Burkhard Becher |

The funders had no role in study design, data collection and interpretation, or the decision to submit the work for publication.

### Author contributions

FM, Conception and design, Acquisition of data, Analysis and interpretation of data, Drafting or revising the article; SJ, RH, Acquisition of data, Analysis and interpretation of data, Drafting or revising the article; LO, BL, Drafting or revising the article, Contributed unpublished essential data or reagents; MH, Acquisition of data, Drafting or revising the article; AW, Conception and design, Drafting or revising the article, Contributed unpublished essential data or reagents; BB, Conception and design, Analysis and interpretation of data, Drafting or revising the article

### Author ORCIDs

Stefanie Joller, http://orcid.org/0000-0002-6906-4892
Burkhard Becher, http://orcid.org/0000-0002-1541-7867

### Ethics

Animal experimentation: All animal experiments were approved by local authorities (Swiss cantonal veterinary office Zurich, KVET license numbers 86/2012, 70/2015, 100/2015 and 68/2013) and performed in strict accordance with the corresponding license.

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
