## [Decision Letter]

Thank you for submitting your work entitled "Signaling via the NFκB-inducing kinase is essential for the function of skin-resident and IL-17 producing γδ T cells" for peer review at *eLife*. Your submission has been favorably evaluated by Tadatsugu Taniguchi (Senior editor), Fiona Powrie (Reviewing editor) and two reviewers.

The reviewers have discussed the reviews with one another and the Reviewing editor has drafted this decision to help you prepare a revised submission.

Mair and colleagues analyzed γδ T cells in NIK-deficient mice and observed two interesting defects:

1) The embryonic development of γ5δ1 epidermal T cells is strongly affected;

2) The IL-17 production capacity of γ4 and γ6 CD27^-^ T cells in spleen and lung is markedly reduced, together with a reduction of Rorc and the receptor for IL-23.

Experiments with a novel conditional knockout crossed with a Cre deleter expressed in lymph node stroma and medullary thymic epithelial cells demonstrated that NIK signaling is required in a cell extrinsic fashion, for the development of γ5. Conditional knockout crossed with a FoxN1-deleter, which acts both in cortical and medullary thymic epithelia showed a defect of IL-17 production capacity of γδ T cells.

This study is interesting, well-organized and performed. It contains a substantial amount of work, is technically sound and advances our understanding of T cell development.

Specific comments:

1) There are reservations regarding the novelty of this study, because of a previous study published in Immunology in 2014 by Eshima et al., which shows a defect of γδ T cells in Nik-deficient mice. This paper is not cited and should be discussed.

2) A major conceptual gap which is neither addressed nor properly discussed is which upstream receptor(s) control NIK activity in CD27^-^ γδ T cells. Is it LTbR? Or RANK? The issue of upstream receptors is also relevant for DETC development since NIK^-/-^ mice have a stronger phenotype than RANK^-/-^ animals. The authors should at a minimum discuss these issues carefully. Another possibility would be to stimulate thymic stroma (from FTOC) with agonists for these receptors and biochemically check NIK activation.

3) Figure 2 shows reduced IL-17 producers in the spleen and lung of NIK^-/-^ mice. What about the dermis – in steady-state and upon psoriasis induction – a model fully established in the Becher lab? Furthermore, how is the disease scored in these animals (if there is an IL-17 phenotype in the dermis)?

4) Figure 4: what is the embryonic THYMIC Vγ5^+^ phenotype (at E15-E16) of the CCL19-conditional NIK^-/-^ mice?

5) Figure 5: in comparison with FoxN1-conditional NIK^-/-^ mice, what is the SPLENIC IL-17 phenotype of CCL19-conditional NIK^-/-^ animals? (This could also be shown in Figure 4).

6) The title is misleading: the effect on skin-resident γδ T cells (DETC) is not functional, but rather developmental/maturation.

7) The authors should cite Ribot et al., J Immunol, 2010 in the fifth paragraph of the Introduction when discussing the role of non-canonical NKkB signalling in γδ T cell biology. This paper shows that CD27 costimulation of CD27^+^ γδ T cells activates non-canonical NF-kB signalling.

8) In the second paragraph of the subsection “In the absence of NIK the development of DETCs is halted in the embryonic thymus”, when describing Figure 1: the difference in numbers of Vγ5^+^ thymocytes is *not* significant. The drop in total γδ thymocytes (Figure 1) must therefore be due to non-Vγ5^+^ cells. This should be qualified. Given the subsequent clear-cut data in Figure 1, the text should clarify a maturation defect in thymic development of DETC precursors, which likely accounts for the skin phenotype (Figure 1).

9) Figure 1: Vγ5^+^ thymocyte development peaks at E15-E16, not at E19 when the analysis was performed. The authors should analyse this earlier ontogeny stage for proper assessment of DETC precursor development (rather than thymic maintenance).

10) Figure 2: did the authors check the Vγ4 versus Vγ6 repertoire of IL-17+ γδ T cells in the lung of NIK^-/-^ and control mice?

11) Discussion: The authors should qualify their conclusion as follows "the impact of non-canonical signaling via NIK on the development of DETC precursors and IL-17-producing γδ T cells stems from signaling within the thymic stroma". They cannot generalize this conclusion to other aspects of γδ T cell development which were not addressed in this study.

---

## [Author Response]

Specific comments:

*1) There are reservations regarding the novelty of this study, because of a previous study published in Immunology in 2014 by Eshima et al., which shows a defect of γδ T cells in Nik-deficient mice. This paper is not cited and should be discussed.*

We apologize for not including the work by Eshima et al. in our original reference list. However, this study investigates the mere presence or absence of γδ T cells in various organs, and has in our opinion the following limitations: First and foremost, bone marrow chimeric mice do not permit the study of γδ T cell development and function, because chimeric mice do not reconstitute all γδ T cell compartments (Haas et al., Immunity, 2012 and Gray et al., JI, 2011). Second, neither in Materials and methods nor in the figure legend (Figure 4) do Eshima et al. state how the absolute numbers of γδ T cells have been measured and calculated. In our hands, systematic quantification of the γδ T cell pool in large numbers of NIK^-/-^ mice never showed such a drastic reduction of absolute γδ T cell counts in spleen, blood or lung. The relative reduction of TCRγδ^+^ cells within the CD3 gate could also be explained by an expansion of CD4^+^ and CD8^+^ T cells. We have included and discussed the study by Eshima et al. in our revised manuscript.

*2) A major conceptual gap which is neither addressed nor properly discussed is which upstream receptor(s) control NIK activity in CD27^-^ γδ T cells. Is it LTbR? Or RANK? The issue of upstream receptors is also relevant for DETC development since NIK^-/-^ mice have a stronger phenotype than RANK^-/-^ animals. The authors should at a minimum discuss these issues carefully. Another possibility would be to stimulate thymic stroma (from FTOC) with agonists for these receptors and biochemically check NIK activation.*

We regret to not have clarified this important point better. Based on our analysis of the conditional NIK allele, NIK signaling neither within γδ T cells nor in αβ T cells is required for the expression of IL17. Instead, only the TEC-specific knockout of NIK led to a loss of IL17 secretion by γδ T cells. Hence, at least for this function of γδ T cells, γδ T cell intrinsic NIK activity is not essential.

We have revised the discussion of this aspect in our revised manuscript in order to state clearly that NIK signaling probably via activation of LTβR and RANK in thymic epithelial cells is required for IL-17 secretion by γδ T cells.

*3) Figure 2 shows reduced IL-17 producers in the spleen and lung of NIK^-/-^ mice. What about the dermis – in steady-state and upon psoriasis induction – a model fully established in the Becher lab? Furthermore, how is the disease scored in these animals (if there is an IL-17 phenotype in the dermis)?*

We have performed additional experiments to analyze the cytokine production by γδ T cells in the dermis, as well as the clinical response of NIK-deficient mice to Aldara-induced skin inflammation. We found that during steady state the functional impairment of NIK-deficient dermal γδ T cells was comparable to the lung γδ T cell compartment. This resulted in a minor change in clinical score after Aldara-induced skin inflammation and a modest albeit not significant reduction of skin thickness. One likely explanation is that innate lymphocytes also contribute to the delivery of IL-17 and IL-22 to the inflamed skin and are likely to compensate for the loss of γδ T cell derived IL-17 (Pantelyushin et al., JCI, 2012).

These additional datasets have been included and discussed in the revised version of the manuscript.

*4) Figure 4: what is the embryonic THYMIC Vγ5^+^ phenotype (at E15-E16) of the CCL19-conditional NIK^-/-^ mice?*

We have attempted those experiments over the past months and observed that the timing of CCL19-Cre mediated recombination of the NIK gene locus in our hands appears too variable to draw definitive experimental conclusions. That is the reason why we decided to target thymic epithelial cells using FoxN1-Cre, which is active earlier and in our hands had a consistent penetrance. We have discussed this aspect in the context of point 5 and amended the Discussion of our manuscript accordingly.

*5) Figure 5: in comparison with FoxN1-conditional NIK^-/-^ mice, what is the SPLENIC IL-17 phenotype of CCL19-conditional NIK^-/-^ animals? (This could also be shown in Figure 4).*

This astute question relates also to point 4. Our analysis of the IL17-production by splenic γδ T cells in CCL19-Cre NIK^flox/flox^ mice showed no difference compared to littermate controls. This is in contrast to our observations with FoxN1-Cre NIK^flox/flox^ animals, and in all likelihood a result of the different time points of Cre expression during development in these two strains: while FoxN1-Cre targets all TECs from an early embryonic stage onwards (Gordon et al., BMC Dev Biol., 2007), CCL19-Cre is active only in maturing mTECs (Onder et al., EJI, 2015). Since the “window” for the development of IL-17-producing γδ T cells is tightly regulated within an early stage in the embryonic thymus (Haas et al., Immunity, 2012), this would explain our observations using these two different Cre lines. In addition, as mentioned above, we have observed variability in the penetrance of the CCL19 transgene, which might also reflect the delicate timing in the development of the thymic medulla (Onder et al., EJI, 2015). We have included and discussed the data obtained from CCL19-Cre NIK^flox/flox^ splenic γδ T cells in the revised manuscript.

*6) The title is misleading: the effect on skin-resident γδ T cells (DETC) is not functional, but rather developmental/maturation.*

We agree and have changed our title to “The NFκB-inducing kinase is essential for the developmental programming of skin-resident and IL-17-producing γδ T cells”.

*7) The authors should cite Ribot et al., J Immunol, 2010 in the fifth paragraph of the Introduction when discussing the role of non-canonical NKkB signalling in γδ T cell biology. This paper shows that CD27 costimulation of CD27^+^ γδ T cells activates non-canonical NF-kB signalling.*

We agree; the work by Ribot et al. has been included in the Introduction as well as the revised Discussion of our manuscript.

*8) In the second paragraph of the subsection “In the absence of NIK the development of DETCs is halted in the embryonic thymus”, when describing Figure 1: the difference in numbers of Vγ5^+^ thymocytes is* not *significant. The drop in total γδ thymocytes (Figure 1) must therefore be due to non-Vγ5^+^ cells. This should be qualified. Given the subsequent clear-cut data in Figure 1, the text should clarify a maturation defect in thymic development of DETC precursors, which likely accounts for the skin phenotype (Figure 1).*

Indeed, the data suggest a maturation defect of DETC precursors, as suggested by the insignificant difference in the numbers of Vγ5+ thymocytes. We have changed the wording in the revised version accordingly and discussed this aspect in the context of our experiments with E17 embryonic thymi (see Figure 1—figure supplement 1 and point 9 below).

*9) Figure 1γ5^+^ thymocyte development peaks at E15-E16, not at E19 when the analysis was performed. The authors should analyse this earlier ontogeny stage for proper assessment of DETC precursor development (rather than thymic maintenance).*

Unfortunately, within the past two months our attempts with timed pregnancies were unsuccessful, and we could not obtain a sufficient number of NIK-deficient E16 embryos. However, in our revised manuscript we have included previously generated data on the development of Vγ5^+^ DETC precursors in E17 thymi (Figure 1—figure supplement 1). We hope that the E17 data, which show similarly impaired development of DETC precursors as on E19, will satisfy this referee. However, we are continuing to attempt analysis of earlier time points. We leave it to the editor to decide whether E17 data are sufficient or whether E16 data are still required. To do so we would require an additional 6-8 weeks in order to set up more timed pregnancies.

*10) Figure 2: did the authors check the Vγ4 versus Vγ6 repertoire of IL-17+ γδ T cells in the lung of NIK^-/-^ and control mice?*

We have performed additional experiments to investigate this and observed that although, as reported, the overall frequency of IL-17-producing γδ T cells in the lung of NIK^-/-^ mice was consistently reduced, the relative proportion of Vγ4^+^ and Vγ6^+^ cells within this pool remained unaltered. We have included this dataset in Figure 2—figure supplement 1.

*11) Discussion: The authors should qualify their conclusion as follows "the impact of non-canonical signaling via NIK on the development of DETC precursors and IL-17-producing γδ T cells stems from signaling within the thymic stroma". They cannot generalize this conclusion to other aspects of γδ T cell development which were not addressed in this study.*

We apologize for this generalization and have rephrased our conclusions based on your suggestion.

In addition to the above mentioned points we have generated new data showing not only that the expression of RORc and IL23R are affected in NIK-deficient CD27^-^γδ T cells, but also the expression of the essential transcription factors Sox13 (Gray et al., Nat Immunol., 2013) and Egr3 (Turchinovic, Immunity, 2011) is deregulated in NIK-deficient CD27^-^γδ T cells. We have incorporated these additional data sets in Figure 3.

Also, as suggested, we have hyphenated “IL-17-producing” as well as “NFkB-inducing” throughout our text. Furthermore, when addressing gene deficient mice, we have replaced “NIK” with the official name of the gene locus, Map3k14.